# Functional Connectivity Increases in Response to High-Definition Transcranial Direct Current Stimulation in Patients with Chronic Disorder of Consciousness

**DOI:** 10.3390/brainsci12081095

**Published:** 2022-08-18

**Authors:** Jinying Han, Chen Chen, Shuang Zheng, Ting Zhou, Shunyin Hu, Xiaoxiang Yan, Changqing Wang, Kai Wang, Yajuan Hu

**Affiliations:** 1Department of Neurology, The First Affiliated Hospital of Anhui Medical University, Hefei 230032, China; 2Collaborative Innovation Center of Neuropsychiatric Disorders and Mental Health, Hefei 230032, China; 3Anhui Province Key Laboratory of Cognition and Neuropsychiatric Disorders, Hefei 230032, China; 4The School of Mental Health and Psychological Sciences, Anhui Medical University, Hefei 230032, China; 5Department of Neurology, The Second Affiliated Hospital of Anhui University of Chinese Medicine, Hefei 230001, China; 6Department of Neurorehabilitation, Hefei Anhua Trauma Rehabilitation Hospital, Hefei 230011, China; 7Institute of Artificial Intelligence, Hefei Comprehensive National Science Center, Hefei 230032, China

**Keywords:** disorders of consciousness (DOCs), Coma Recovery Scale-Revised (CRS-R), high-definition transcranial direct current stimulation (HD-tDCS), phase locking value (PLV)

## Abstract

**Highlights:**

Functional connectivity induced by HD-tDCS in DLPFC has different trends in CRS-R score improvers and non-improvers.An increase in theta PLV in the left frontal–parietooccipital region was significantly associated with CRS-R changes.DOC patients with increased PLV of the alpha band in the intra-bifrontal region have a better prognosis than those without.

**Abstract:**

High-definition transcranial direct current stimulation (HD-tDCS) has been shown to play an important role in improving consciousness in patients with disorders of consciousness (DOCs), but its neuroelectrophysiological evidence is still lacking. To better explain the electrophysiological mechanisms of the effects of HD-tDCS on patients with DOCs, 22 DOC patients underwent 10 anodal HD-tDCS sessions of the left dorsolateral prefrontal cortex (DLPFC). This study used the Coma Recovery Scale-Revised (CRS-R) to assess the level of consciousness in DOC patients. According to whether the CRS-R score increased before and after stimulation, DOC patients were divided into a responsive group and a non-responsive group. By comparing the differences in resting-state EEG functional connectivity between different frequency bands and brain regions, as well as the relationship between functional connectivity values and clinical scores, the electrophysiological mechanism of the clinical effects of HD-tDCS was further explored. The change of the phase locking value (PLV) on the theta frequency band in the left frontal–parietooccipital region was positively correlated with the change in the CRS-R scores. As the number of interventions increased, we observed that in the responsive group, the change in PLV showed an upward trend, and the increase in the PLV appeared in the left frontal–parietooccipital region at 4–8 Hz and in the intra-bifrontal region at 8–13 Hz. In the non-responsive group, although the CRS-R scores did not change after stimulation, the PLV showed a downward trend, and the decrease in the PLV appeared in the intra-bifrontal region at 8–13 Hz. In addition, at the three-month follow-up, patients with increased PLV in the intra-bifrontal region at 8–13 Hz after repeated HD-tDCS stimulation had better outcomes than those without. Repeated anodal stimulation of the left DLPFC with HD-tDCS resulted in improved consciousness in some patients with DOCs. The increase in functional connectivity in the brain regions may be associated with the improvement of related awareness after HD-tDCS and may be a predictor of better long-term outcomes.

## 1. Introduction

Brain damage caused by various reasons may impair the production and maintenance of consciousness, thereby affecting behavioral output [1,2]. Common causes of disturbance of consciousness include traumatic brain injury, hypoxic-ischemic encephalopathy, cerebrovascular disease, central nervous system infection, and poisoning. When the disturbance of consciousness exceeds 28 days, it is defined as a chronic disturbance of consciousness (prolonged DOC) and includes vegetative state (VS) or minimally conscious state (MCS) patients [3]. VS patients lack behavioral evidence of self and environmental awareness, but arousal is preserved, eyes can be opened, and a sleep–wake cycle exists. In contrast, MCS is a disorder of altered consciousness characterized by minimal but definite behavioral evidence of self or environmental awareness. It exhibits repeatable but inconsistent signs of consciousness such as command following, visual pursuit, and localization to noxious stimuli [4]. In the clinic, it is a challenge to evaluate and treat patients with DOCs because of the huge costs of prolonged intensive care.

In recent years, transcranial direct current stimulation (tDCS) has been increasingly demonstrated to play an important role in the recovery of impaired consciousness in patients with DOCs [5,6,7]. High-definition transcranial direct current stimulation (HD-tDCS) is a form of non-invasive neurostimulation that modulates cortical excitability through the action of weak polarizing currents [8]. Compared with tDCS, HD-tDCS is more precise and can lead to focal neuromodulation and specific behavioral changes [9,10]. It delivers low-intensity direct current (1−2 mA) to areas of the cerebral cortex and has been proven effective in many studies related to disorders of consciousness [6]. Anodal stimulation increases cortical excitability, while cathodal stimulation decreases cortical excitability. tDCS has obvious positive effects and has the advantages of portability, low cost, and no obvious side effects [11]. Studies have shown that tDCS has a significant difference in the treatment effect for VS and MCS patients [12]. A study has shown that following multifocal frontoparietal tDCS, the electroencephalographic (EEG) complexity significantly increased in low frequency bands (1–8 Hz), while CRS-R total score improvement was associated with decreased baseline complexity in those bands [13]. A previous study also showed increased baseline spatial connectivity and higher network centrality in the theta band in DLPFC tDCS-responders as compared to non-responders [14]. Despite accumulating evidence that the use of tDCS devices can enhance the responsiveness of some patients with DOCs [15], the effects of tDCS on EEG measures of complexity are still not fully understood. From a new perspective of whether DOC patients respond to tDCS, this study focused on the differences in EEG functional connectivity between the two groups, which can better explain the possible changes in brain neural circuits in patients with clinical improvement. This study aims to provide more useful clues for clarifying the neurophysiological mechanism of tDCS in patients with DOCs.

There are also many tDCS intervention target options for DOC patients. tDCS can be applied over the left DLPFC, posterior parietal cortex, and left primary sensorimotor cortex [6,16,17]. In clinical studies, these above tDCS stimulation targets have been reported to have intervention effects on DOC patients. The DLPFC is considered to be one of the key brain regions for top-down control, and it has been shown that increasing extrastriate neural activity modulates bottom-up processing, thereby enhancing attention to stimuli [18,19,20]. In fact, the prefrontal lobe is a common stimulation target for non-invasive neurostimulation protocols, which can well assess and regulate the level of consciousness. After applying tDCS intervention on the left DLPFC, findings showed significant improvement in the behavior of MCS patients [21,22]. Data from one follow-up showed that after 12 months of follow-up, 13/30 MCS patients exhibited signs of consciousness associated with anodal tDCS [6]. Thus, selective tDCS stimulation of prefrontal regions produces distinct clinical behavioral and electrophysiological effects [23]. Therefore, tDCS stimulation of the left DLPFC has important value for the recovery of consciousness in DOC patients.

The regulation of consciousness is complex, involving multiple brain regions and brain networks. It is generally accepted that two networks have been identified as potential mediators of consciousness, including the default mode network (DMN) and the frontoparietal network (FPN). The DMN is functionally related to internal consciousness, while the FPN is mainly related to the processing of external stimuli [24,25]. The DMN, which reflects the internal states related to alertness and self-related processes, is involved in cognitive functions related to internal processing and external input. The frontoparietal connection is important to the thalamocortical network, which regulates cortical states and behaviors, including perception, learning, and cognition, and is a component of consciousness [26]. Studies have found that the frontoparietal functional connectivity of patients with DOCs is generally interrupted [27,28]. The difference in the preservation of the frontoparietal network connection of DOC patients may be an important reason for the difference in the level of consciousness [29,30].

In the present study, we employed resting-state EEG to assess cortical excitability in DOC patients treated with HD-tDCS, by comparing the response group (RE) and non-response group (N-RE). Specifically, the subjects were divided into an RE group and an N-RE group based on the changes in Coma Recovery Scale-Revised (CRS-R) scores before and after stimulation. The aim of the study was to demonstrate the effectiveness of HD-tDCS in DOC patients and explore the neuroelectrophysiological mechanism of brain network functional connectivity changes caused by HD-tDCS over the left DLPFC, which will provide more useful evidence for clinical intervention in patients with DOCs.

## 2. Experimental Procedures

### 2.1. Patients

From September 2020 to December 2021, DOC patients from the Department of Neurology of the First Affiliated Hospital of Anhui Medical University (Hefei, China), the Department of Neurology of the Second Affiliated Hospital of Anhui University of Chinese Medicine (Hefei, China), and the Department of Neurorehabilitation of Hefei Anhua Trauma Rehabilitation Hospital (Hefei, China) were included in this study. We recruited a total of 22 DOC patients in this study, including 16 males and 6 females, with an average age of 54.45 ± 13.44 years (see Table 1 for details). The average duration of consciousness impairment for all patients was (81.59 ± 70.91) days. The etiology of the DOCs was trauma (4 patients), hypoxic-ischemic encephalopathy (5 patients), cerebral hemorrhage (8 patients), cerebral infarction (4 patients), and fulminant encephalomyelitis (1 patient). Inclusion criteria were: (1) patients diagnosed as VS/MCS based on CRS-R assessment and between 18 and 75 years old; (2) maintained stable vital signs; (3) no use of neuromuscular blocking agents and sedatives in the 24 h prior to the study; (4) no improvement in consciousness was observed in all subjects within 1 week prior to the start of the study. Examples of exclusion criteria for this study are as follows: (1) subjects with severe neurocognitive degenerative disease; (2) head metal implant; (3) previous craniotomy; (4) previous history of epilepsy; (5) past transcranial electrical stimulation or transcranial magnetic stimulation. All patients used drugs and rehabilitation as usual during the study. Written informed consent from relatives was required for all patients participating in this study. This study has been approved by the Ethics Committee of the First Affiliated Hospital of Anhui Medical University.

### 2.2. Behavior

The CRS-R has been proven to have good reliability and validity and has been widely used in the behavioral evaluation of DOC patients [31]. It consists of 23 items arranged in levels, including six subscales used to assess auditory, visual, motor, verbal, communication, and arousal functions. The score is based on whether there is a behavioral response to the stimulus presented in a standardized manner, which ranges from 0 to 23. Higher scores indicate higher neurological function and better prognosis [32,33]. In recent years, MCS has been divided into two subgroups according to the complexity of the patients’ behaviors: “MCS plus” (MCS+) and “MCS minus” (MCS−). MCS+ refers to patients with high behavioral responses, such as executable commands, expression of intelligible language, and responses with gestures or verbally stating “yes” or “no”. MCS− refers to patients with low behavioral responses and non-reflexive movements to stimulus localization and visual cues [34]. When MCS patients exhibit functional communication or can use two different objects they are considered to be in an exit in a minimally conscious state (EMCS) [35]. CRS-R scores were assessed by two trained physicians before stimulation, after the first stimulation, after treatment, and at 3-month follow-up after treatment. When the two doctors scored differently, the other doctor scored again. According to whether the CRS-R score increased before and after stimulation, the patients were divided into the RE group and the N-RE group in this study.

### 2.3. Stimulation Protocol

The stimulation protocol was performed by a direct current stimulator (Neuroelectrics, Barcelona, Spain) consisting of one anodal electrode and four cathodal electrodes. The anodal electrode was placed over the left DLPFC (F3 in the 10–20 international system EEG placement), and the four cathodal electrodes (AFz, FCz, F7, and C5 in the 10–20 international system EEG placement) were placed around the anodal electrode to form a current loop. The current was ramped up to 2 mA over 30 s before the start of the experiment. Then, the current was maintained for 20 min, and slowly ramped down over 30 s after stimulation. As shown in Figure 1, all patients received ten HD-tDCS sessions for five consecutive days, and the patients’ CRS-R scores and EEGs were recorded before the experiment (T0), after a single HD-tDCS session (T1), and after the final treatment (T2).

### 2.4. EEG Recordings and Pre-Processing

EEG was recorded for at least 6 min using 19 channels (EEG-1200C, Nihon Kohden, Shinjuku, Japan), and Ag/AgCl pin electrodes were used with a sampling rate of 200 Hz. The 19 channels were Fp1, Fp2, F3, F4, F7, F8, Fz, Cz, C3, C4, T3, T4, T5, T6, Pz, P3, P4, O1, and O2. The skin–electrode impedance is required to be kept below 5 kΩ during EEG acquisition. EEG pre-processing was performed using EEGLAB software version 13.0 b running on a MATLAB environment (Version 2013b, MathWorks Inc., Natick, MA, USA). The EEG signal was band filtered between 0.1 and 40 Hz with a notch filter of 48–52 Hz [36]. Independent component analysis was used to identify and remove artifact-relevant components, mainly including eye movements and muscle activation [37]. The selected artifact-free epochs were averaged with reference and rejection epochs when exceeding ±150 uv [38].

### 2.5. EEG Analysis of the Phase Locking Value (PLV)

We divided the brain into four cortical regions: left frontal, right frontal, central, and parietooccipital regions [39,40]. As shown in Figure 2, the left frontal region included electrodes Fp1, F3, and F7; the right frontal region included electrodes Fp2, F4, and F8; the central region included electrodes Cz, C3, and C4; and the parietooccipital region included electrodes Pz, P3, P4, O1, and O2. Research has shown that theta and alpha bands are associated with working memory processing and improved consciousness [14,41,42,43]. Previous results from our team showed that changes in theta power and alpha power were positively correlated with changes in CRS-R scores [7]. Meanwhile, it has been found that anodal tDCS can induce changes in the frontoparietal network and the frontal part of the default mode network [44,45]. Therefore, data were calculated in the following frequency bands: theta (4–8 Hz) and alpha (8–13 Hz) [46]. The left frontal–parietooccipital region PLV was calculated using pairwise electrodes from the left frontal and parietooccipital regions. The intra-bifrontal PLV was also calculated. The connectivity between paired channels in each frequency band was computed using phase synchronization, which was described as follows: for each epoch EEG signal, the instantaneous phases ϕx(t) and ϕy(t) of the paired channels were evaluated based on the Hilbert transform. Then, the phase difference was defined by
Δ*ϕ_xy_* (t) = ϕ_x_ (t) − ϕ_y_ (t).(1)

Several indices based on short-term phase differences can be used to indicate the phase synchronization between two series [47]. This study applied the PLV based on the circular variance of the phase difference and obtained
(2)$${\mathrm{PLV}}_{\mathrm{xy}}=\frac{1}{N}|\sum_{t=1}^{N}e^{j\Delta \varphi xy\left(t\right)}|$$

This measure of the PLV varied between 0 and 1, and the computation involved no parameter selection. Therefore, the synchronization can be described by each element of the phase synchronization matrix C and PLV_xy_.

### 2.6. Statistical Analysis

Baseline characteristics between the RE and N-RE groups were compared using the χ^2^ test for categorical variables and the t-test or Mann–Whitney U test for continuous variables. Three-month follow-up outcomes were compared using Fisher’s exact test. Two-way repeated ANOVA with “group” as a between-subject factor and “time” as a within-subject factor was used to analyze the changes in PLV at different time points between the RE and N-RE groups. The assumption of sphericity was assessed using Mauchly’s test before repeated measures ANOVA. When the hypothesis was rejected, the Greenhouse–Geisser correction was used to adjust the degrees of freedom. Post hoc analyses were performed using Sidak’s multiple comparison test. FDR correction was used to correct for multiple comparisons. Finally, Spearman’s rank test correlation analysis was used to analyze the relationship between PLV changes and the increase in CRS-R scores of the DOC patients. *p* < 0.05 was considered statistically significant.

## 3. Results

### 3.1. Demographic and Clinical Behavioral Outcomes

The entire research flow chart is shown in Figure 3. We included a total of 22 subjects (9 MCS, 13 VS) and analyzed the CRS-R scores before and after stimulation. No specific side effects were seen after HD-tDCS treatment, such as itching, burning, or epilepsy. Increases in CRS-R scores were observed in four patients after a single HD-tDCS stimulation but were not statistically significant (t = −2.017, *p* = 0.057). After five consecutive days of HD-tDCS stimulation, patients’ CRS-R scores were significantly higher than before stimulation (t = −3.512, *p* = 0.002, Figure 4A). Furthermore, we observed significant improvement in all six subscales of the CRS-R after repeated stimulation (Figure 4B), including auditory (t = −3.480, *p* = 0.002), visual (t = −2.569, *p* = 0.018), motor (t = −2.614, *p* = 0.016), verbal (t = −2.347, *p* = 0.029), communication (t = −2.160, *p* = 0.042), and arousal functions (t = −2.160, *p* = 0.042).

Three of the subjects did not participate in the analysis of EEG indicators due to EEG quality problems. A total of 19 individuals were divided into an RE group and an N-RE group according to whether there was an increase in CRS-R scores after repeated HD-tDCS. The RE group included seven MCS and four VS patients, including eight males and three females. In the N-RE group, there was one MCS and seven VS patients, including five males and three females. The mean age in the RE group was 57.81 ± 9.51 years, and the mean onset time was 74.81 ± 78.98 days. The mean age in the N-RE group was 49.5 ± 16.68 years, and the mean onset time was 85.62 ± 60.39 days. There were no significant differences in gender (z = −0.461, *p* = 0.717), age (t = −1.382, *p* = 0.185), onset time (z = −1.033, *p* = 0.302), or etiology (χ^2^ = 5.807, *p* = 0.191) between the RE and N-RE groups. In the RE group, etiology included trauma, hypoxic-ischemic encephalopathy, cerebral hemorrhage, and cerebral infarction, whereas in the N-RE group, etiologies included fulminant encephalomyelitis, hypoxic-ischemic encephalopathy, cerebral hemorrhage, and cerebral infarction.

### 3.2. Electroencephalographic Results

We observed a significant interaction effect between time (T0, T1, and T2) and group (RE and N-RE) on PLV at 4–8 Hz in the left frontal–parietooccipital region (F_(2,34)_ = 7.468, *p* = 0.004). No significant difference was observed in PLV at 4–8 Hz in the RE group (from 0.48 to 0.52, t = 2.775, *p* = 0.1556, effect size = 0.83; 95% CI, 0.12–1.51, Figure 5A) and the N-RE group (from 0.45 to 0.44, t = 0.4443, *p* = 0.9006, effect size = 0.15; 95% CI, −0.54–0.84) at T1. Different results were observed after repeated stimulation (T2). Specifically, there was an improvement in PLV among patients in the RE group (from 0.48 to 0.56, t = 4.525, *p* = 0.0088, effect size = 1.36; 95% CI, 0.51–2.18, Figure 5A) but not the N-RE group (from 0.45 to 0.43, t = 0.8578, *p* = 0.6629, effect size = 0.30; 95% CI, −0.41–1.00). Finally, the RE group displayed higher PLV than the N-RE group at T1 (mean difference = 0.07, t = 4.146, *p* = 0.008, effect size = 1.92; 95% CI, 0.79–3.02) and T2 (mean difference =0.12, t = 4.023, *p* = 0.0058, effect size = 1.86; 95% CI, 0.73–2.96).

Meanwhile, in the intra-bifrontal region at 8–13 Hz, there was an interaction effect between time (T0, T1, and T2) and group (RE and N-RE) on PLV (F_(2,34)_ = 8.822, *p* = 0.0032).The PLV at T2 was significant increased in the RE group (from 0.44 to 0.56, t =2.671, *p* = 0.0481, effect size = 0.80; 95% CI, 0.10–1.47, Figure 5B) and significant decreased in the N-RE group (from 0.38 to 0.34, t =4.004, *p* = 0.0412, effect size = 1.41; 95% CI, 0.38–2.39, Figure 5B), but no significant changes were observed in both the RE (from 0.44 to 0.45, t = 0.613, *p* = 0.8557, effect size = 0.18; 95% CI, −0.41–0.77) and N-RE group (from 0.38 to 0.36, t = 1.532, *p* = 0.9006, effect size = 0.54; 95% CI, −0.22–1.27) at T1. Similarly, the RE group PLV was significantly higher than the N-RE group PLV at T1 (mean difference = 0.08, t =3.234, *p* = 0.0346, effect size =1.50; 95% CI, 0.42–2.54) and T2 (mean difference = 0.22, t = 6.246, *p* = 0.0004, effect size = 2.90; 95% CI, 1.45–4.30).

In the left frontal–parietooccipital region, we found that the variation of theta functional connectivity had a significant positive correlation with the variation of CRS-R between T2 and T0 (r = 0.694, *p* = 0.001, Figure 6) for all the patients, but not between T1 and T0 (r = 0.069, *p* = 0.779). We followed all patients for three months and divided the outcomes into EMCS and others. According to whether the PLV at 8–13 Hz in the intra-bifrontal region increased after repeated stimulation, the patients were divided into a PLV increased group and a non-increased group. We found that DOC patients with increased PLV had significantly better long-term outcomes than patients without increased PLV (*p* = 0.018). In addition, we observed improvement in CRS-R scores after HD-tDCS stimulation in four patients diagnosed with VS. Figure 7 and Figure 8 show the PLV values of the RE group, the N-RE group, and a VS patient (NO.8) at three time points in the 4–8 Hz and 8–13 Hz frequency bands, respectively.

## 4. Discussion

To determine the effect of HD-tDCS on the functional connectivity of DOC patients, we measured resting-state EEG before and after HD-tDCS stimulation. Specifically, we analyzed the changes in functional connectivity of different brain regions and frequency bands in the RE and N-RE groups before and after anodal HD-tDCS treatment, thereby investigating the possible neural pathways that HD-tDCS regulates in the cerebral cortex. After HD-tDCS treatment, an increase in PLVs was detected in the RE group in the left frontal–parietooccipital region, while a decrease in PLVs was detected in the N-RE group. In addition, in the left frontal–parietooccipital region, we also found a positive correlation between changes in PLVs in the theta frequency band and CRS-R score changes before and after HD-tDCS intervention. These findings suggested that HD-tDCS effectively modulates the functional connectivity changes between different brain regions, especially in the left frontal–parietooccipital region and the intra-bifrontal region, ultimately leading to different clinical outcomes in patients with DOCs.

In the RE group, we observed increased functional connectivity at 8–13 Hz in the intra-bifrontal region. A previous study also found that alpha functional connectivity increased within the frontal lobe after tDCS [39]. At the same time, there is evidence indicating that alpha functional connectivity is closely related to the degree of consciousness in patients with DOCs, which is also believed in the use of sedatives [48,49]. In addition to causing changes in stimulated brain regions, tDCS can also cause changes in distant regions. An fMRI study showed that tDCS caused increased synergistic activation of different frontal lobes, revealing that tDCS on the left DLPFC could affect the frontal part of the DMN [45]. In our study, there was an increased functional connectivity within the frontal lobe, which further confirmed that tDCS improved the state of consciousness in patients with DOCs and that the improvement may be achieved by the coactivation of the frontal region of the DMN in a neuroelectrophysiological perspective. In addition, during the three-month follow-up, we found that DOC patients whose functional connectivity values increased in the intra-bifrontal region tended to have better clinical performance after HD-tDCS stimulation, while DOC patients who showed a decreasing trend tended to have greater difficulty in recovery from impairment of consciousness. Perhaps functional connectivity changes induced by HD-tDCS intervention may have a certain degree of value in predicting the prognosis of patients with DOCs.

Recently, a series of studies have found that the theta band appeared to be involved in consciousness, top-down activation, and information transfer between memory systems [14,41,42,50]. It has been shown that tDCS on the left DLPFC can cause an increase in theta functional connectivity in the frontoparietal region [22]. Similarly, in this study, with the increase in stimulations, we found a significant increase in functional connectivity in the left frontal–parietooccipital region at 4–8 Hz in the RE group, but not in the N-RE group. A study of prefrontal tDCS on patients with DOCs indicated that responders showed increases in power and long-range corticocortical functional connectivity in the theta–alpha band, compared with non-responders [51]. Improvements in consciousness may be related to changes in brain networks [52,53]. tDCS on the prefrontal region could modulate large-scale patterns of resting-state connectivity in the human brain by inducing functional changes based on a resting-state fMRI analysis. These effects were detectable in three resting-state networks, namely the DMN and the left and right FPN [45]. The FPN was believed to have an important role in external awareness [54]. Meanwhile, another resting-state fMRI study showed that tDCS stimulation of the DLPFC can promote recovery of consciousness in MCS patients with high connectivity in external control networks [55]. In addition, a study considered that tDCS could modulate the propagation of alpha and beta waves between forebrain and hindbrain regions in the DLPFC via long-distance frontoparietal connections when certain cognitive functions were preserved in DOC patients [56]. These results highlight the important role of long-range frontoparietal connectivity in consciousness and show the potential therapeutic utility of tDCS [56]. Consistent with previous studies, we observed a significant positive correlation between the changes in PLV in the left frontal–parietooccipital region at 4–8 Hz after repeated electrical stimulation and the changes in CRS-R [14]. Therefore, we believe that HD-tDCS intervention on the left DLPFC leads to clinical behavioral changes in DOC patients by acting on the extrinsic consciousness network.

Compared with healthy individuals, patients with DOCs lack structured long-range functional connectivity [57]. Compared with MCS patients, VS patients may have less preservation of corticocortical and corticothalamic interactions and reduced information integration ability of various networks due to anatomical damage [58,59]. However, there is still a certain misdiagnosis rate in distinguishing between MCS and VS based on the clinical behavior scale, even reaching 43% [60]. A study based on fMRI data collected with the tennis and spatial navigation imagery task determined that two patients diagnosed with VS based on CRS-R scores had the ability to follow commands [61]. In our study, we observed improvements in CRS-R scores and PLV increased even within several VS patients (4/13) after HD-tDCS intervention. We suggest that VS patients diagnosed based on the CRS-R score may also retain residual brain networks and have the potential to improve some degree of consciousness. Since MCS has been shown to have better clinical improvement compared to VS in previous studies [6], it is extremely important to distinguish MCS from VS. Functional connectivity between auditory and visual cortices was found to be the most sensitive feature to accurately delineate patients as being in an MCS or VS. Meanwhile, changes in coherence elicited by anodal tDCS in the prefrontal cortex also emerged as a tool to differentiate an MCS from a VS [22]. In the future, we can further explore methods to distinguish MCS from VS based on functional connectivity responsiveness elicited by tDCS and to assess the potential plasticity of residual brain networks.

This study also has certain limitations, such as the small sample size, which may have reduced the statistical power. The etiology of DOCs is varied, and chronic DOC is defined as greater than 28 days with a large time span of illness. The small sample size cannot further explore the mechanism and influencing factors of consciousness improvement according to the etiology and duration of onset. In future studies, we will further expand the sample size and analyze the response of different types of patients to HD-tDCS from the perspective of etiology and duration of onset. Moreover, this study only performed anodal HD-tDCS intervention on all DOC patients, not a randomized controlled study. The increase in CRS-R scores may be the result of spontaneous recovery. Due to the lack of a sham group, it cannot be well demonstrated that the effect of tDCS directly led to the improvement in clinical scores and changes in EEG functional connectivity. Therefore, a sham group should be added and compared with the real HD-tDCS treatment. Finally, the analysis of brain regions by 19-channel EEG is relatively rough, and high-density EEG will be collected in the future for further exploration.

## 5. Conclusions

In this study, by comparing the resting-state EEG data of the RE and N-RE groups, we found significant differences in the left frontal–parietooccipital region functional connectivity before and after HD-tDCS intervention. HD-tDCS improves consciousness by affecting the functional connectivity of brain regions in DOC patients. Taken together, the evidence indicates that functional connectivity can be used to predict the long-term prognosis of patients.

## Figures and Tables

**Figure 1 brainsci-12-01095-f001:**
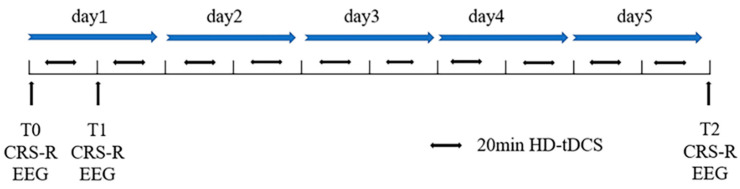
Stimulation Protocol.

**Figure 2 brainsci-12-01095-f002:**
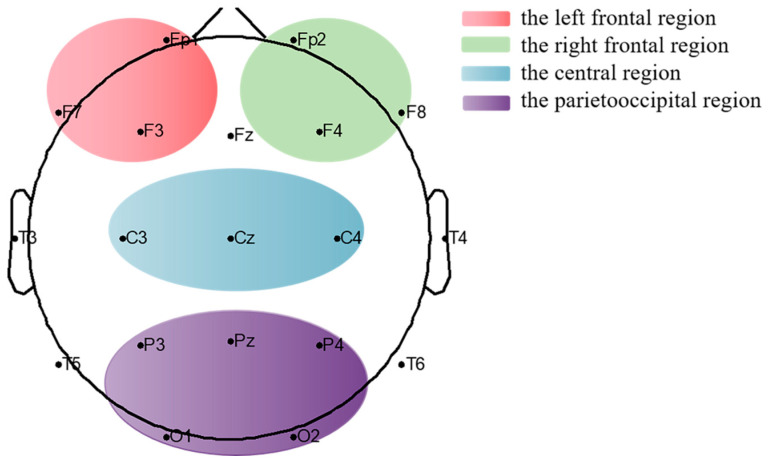
Differently defined regions of the brain.

**Figure 3 brainsci-12-01095-f003:**
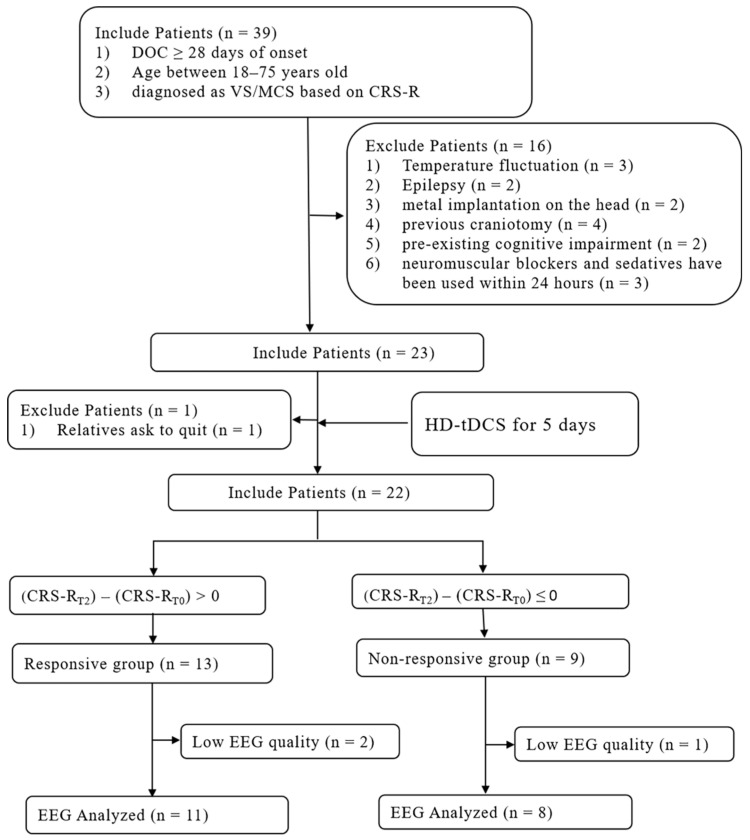
Flow chart of the study.

**Figure 4 brainsci-12-01095-f004:**
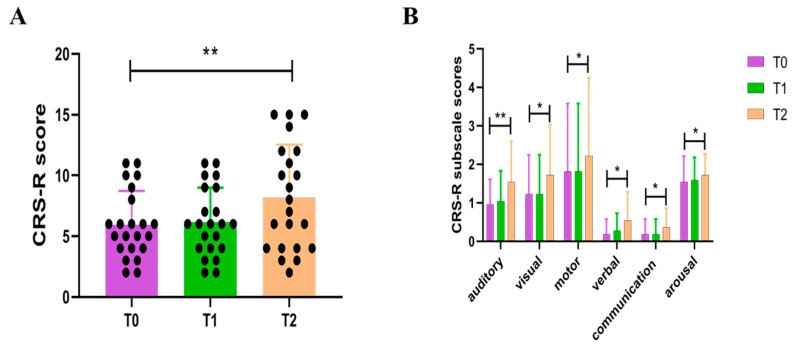
Changes in CRS-R scores at different time points (**A**) and CRS-R subscale scores (**B**). CRS-R, Coma Recovery Scale-Revised; T0, before the experiment; T1, after a single session of HD-tDCS; T2, after the treatment of 5 days. * *p* < 0.05. ** *p* < 0.01.

**Figure 5 brainsci-12-01095-f005:**
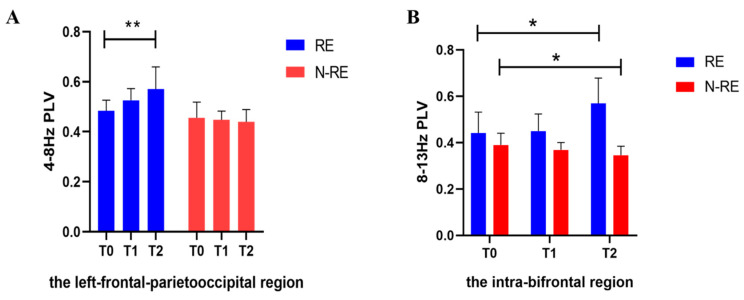
With the increase of stimulation, the PLV of the RE and the N-RE groups showed different changes (**A**) in the left-frontal-parietooccipital region at 4–8 Hz, (**B**) in the intra-bifrontal region at 8–13 Hz; PLV, phase locking value; RE, responsive group; N-RE, non-responsive group; T0, before the experiment; T1, after a single HD-tDCS session; T2, after the treatment of 5 days. * *p* < 0.05. ** *p* < 0.01.

**Figure 6 brainsci-12-01095-f006:**
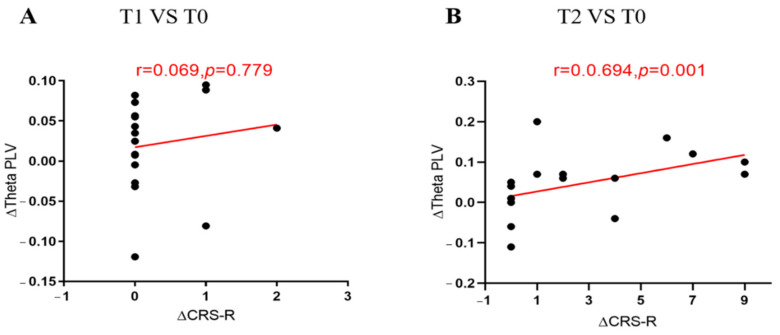
ΔCRS-R, the increases in the CRS-R scores at T2/T1 compared with T0; ΔPLV, the increases in PLV at T2/T1 compared with T0; Comparison of T1 and T0 in the left-frontal-parietooccipital region at 4–8 Hz for all patients (**A**); Comparison of T2 and T0 in the left-frontal-parietooccipital region at 4–8 Hz for all patients (**B**); T0, before the experiment; T1, after a single HD-tDCS session; T2, after the treatment of 5 days. CRS-R, Coma Recovery Scale-Revised; PLV, phase locking value.

**Figure 7 brainsci-12-01095-f007:**
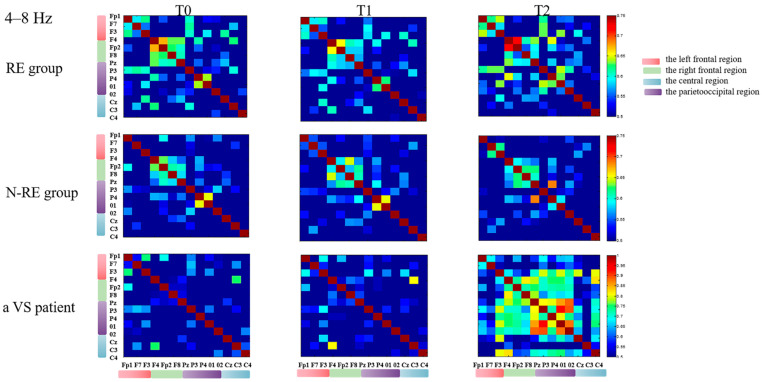
The PLV of the RE group, N-RE group and a VS patient (NO.8) at different times in each channel at 4–8 Hz. RE, responsive group; N-RE, non-responsive group; T0, before the experiment; T1, after a single HD-tDCS session; T2, after the treatment of 5 days. PLV, phase locking value.

**Figure 8 brainsci-12-01095-f008:**
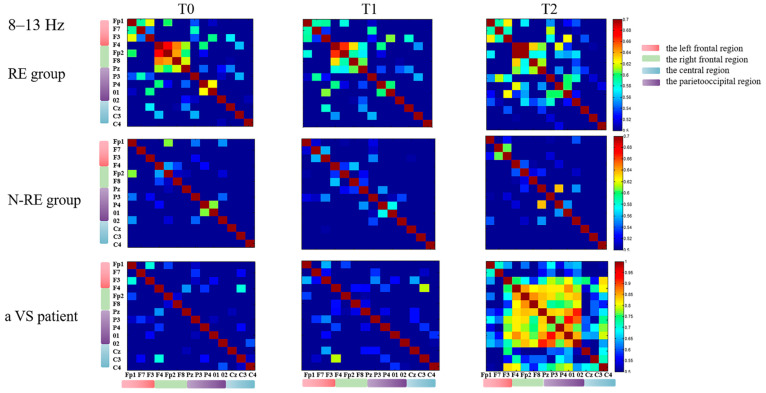
The PLV of the RE group, N-RE group and a VS patient (NO.8) at different times in each channel at 8–13 Hz. RE, responsive group; N-RE, non-responsive group; T0, before the experiment; T1, after a single HD-tDCS session; T2, after the treatment of 5 days. PLV, phase locking value.

**Table 1 brainsci-12-01095-t001:** Participants’ demographic characteristics.

ID	Sex	Age	Etiology	Days Post- Injury	T0 (CRS-R) (A-Vi-M-Ve-C-Ar)	T0—Clinical Diagnosis	T1 (CRS-R) (A-Vi-M-Ve-C-Ar)	T1—Clinical Diagnosis	T2 (CRS-R) (A-Vi-M-Ve-C-Ar)	T2—Clinical Diagnosis	Follow-Up at 3 Months (CRS-R) (A-Vi-M-Ve-C-Ar)	Follow-Up at 3 Months—Clinical Diagnosis
RE1	M	52	Trauma	84	11(1/3/4/1/1/1)	MCS+	11(1/3/4/1/1/1)	MCS+	12(2/3/4/1/1/1)	MCS+	18(3/4/6/1/1/3)	EMCS
RE2	F	49	HIE	30	6(1/1/3/0/0/1)	MCS-	7(2/1/3/0/0/1)	MCS-	15(3/3/5/2/1/1)	MCS+	20(3/4/5/3/2/3)	EMCS
RE3	M	53	Trauma	34	5(0/0/5/0/0/0)	MCS-	7(0/0/5/1/0/1)	MCS-	14(3/2/5/2/1/1)	MCS+	23(4/5/6/3/2/3)	EMCS
RE4	F	74	Hemorrhage	101	11(2/3/3/0/1/2)	MCS+	11(2/3/3/0/1/2)	MCS+	15(3/4/5/0/1/2)	MCS+	/	Dead
RE5	M	49	Hemorrhage	50	5(1/1/1/0/0/2)	VS	6(1/1/1/1/0/2)	VS	7(1/1/2/1/0/2)	VS	8(1/2/2/1/0/2)	VS
RE6	M	55	Trauma	302	6(1/2/1/0/0/2)	VS	6(1/2/1/0/0/2)	VS	8(2/3/1/0/0/2)	MCS-	8(2/3/1/0/0/2)	MCS-
RE7	M	72	Cerebral infarction	42	5(1/3/0/0/0/1)	MCS-	5(1/3/0/0/0/1)	MCS-	9(2/3/0/2/1/1)	MCS+	13(3/3/0/3/1/3)	MCS+
RE8	M	47	Hemorrhage	29	6(1/1/2/1/0/1)	VS	6(1/1/2/1/0/1)	VS	12(2/3/3/1/1/2)	MCS+	14(2/4/3/1/1/3)	MCS+
RE9	M	58	Trauma	53	9(2/3/2/0/0/2)	MCS-	9(2/3/2/0/0/2)	MCS-	10(2/3/2/1/0/2)	MCS-	22(4/4/6/3/2/3)	EMCS
RE10	F	68	Hemorrhage	30	8(2/1/3/0/0/2)	MCS-	9(3/1/3/0/0/2)	MCS+	15(3/4/5/0/0/3)	MCS+	18(3/4/5/2/1/3)	MCS+
RE11	M	59	Cerebral infarction	68	5(2/1/0/0/0/2)	VS	5(2/1/0/0/0/2)	VS	6(3/1/0/0/0/2)	MCS+	15(4/3/5/0/1/2)	MCS+
RE12 #	M	37	HIE	35	10(1/2/5/0/1/1)	MCS+	10(1/2/5/0/1/1)	MCS+	11(2/2/5/0/1/1)	MCS+	18(3/5/6/0/2/2)	EMCS
RE13 #	M	72	Cerebral infarction	200	2(1/1/0/0/0/0)	VS	2(1/1/0/0/0/0)	VS	4(1/1/1/0/0/1)	VS	5(1/2/1/0/0/1)	VS
N-RE1	M	54	Hemorrhage	73	6(1/1/2/0/0/2)	VS	6(1/1/2/0/0/2)	VS	6(1/1/2/0/0/2)	VS	6(1/1/2/0/0/2)	VS
N-RE2	M	56	HIE	41	2(0/0/0/0/0/2)	VS	2(0/0/0/0/0/2)	VS	2(0/0/0/0/0/2)	VS	4(0/0/1/1/0/2)	VS
N-RE3	F	39	HIE	128	4(0/0/2/0/0/2)	VS	4(0/0/2/0/0/2)	VS	4(0/0/2/0/0/2)	VS	5(1/0/2/0/0/2)	VS
N-RE4	M	18	Disseminated cerebrospinalmeningits	48	4(1/1/0/0/0/2)	VS	4(1/1/0/0/0/2)	VS	4(1/1/0/0/0/2)	VS	7(2/2/0/1/0/2)	MCS-
N-RE5	M	56	Hemorrhage	88	3(0/0/1/0/0/2)	VS	3(0/0/1/0/0/2)	VS	3(0/0/1/0/0/2)	VS	5(1/1/1/0/0/2)	VS
N-RE6	M	64	Hemorrhage	34	10(1/1/5/0/1/2)	MCS-	10(1/1/5/0/1/2)	MCS-	10(1/1/5/0/1/2)	MCS-	12(2/1/5/0/1/3)	MCS+
N-RE7	F	70	Cerebral infarction	58	4(1/1/0/1/0/1)	VS	4(1/1/0/1/0/1)	VS	4(1/1/0/1/0/1)	VS	7(1/1/0/2/1/2)	MCS-
N-RE8	F	39	HIE	215	3(0/0/0/1/0/2)	VS	3(0/0/0/1/0/2)	VS	3(0/0/0/1/0/2)	VS	4(1/0/0/1/0/2)	VS
N-RE9 #	M	57	Hemorrhage	52	6(1/1/2/0/0/2)	VS	6(1/1/2/0/0/2)	VS	6(1/1/2/0/0/2)	VS	14(2/2/4/2/1/3)	MCS+

RE, responsive group; N-RE, non-responsive group; HIE, hypoxic-ischemic encephalopathy; VS, vegetative state; MCS+, minimally conscious state plus; MCS-, minimally conscious state minus; EMCS, exit in a minimally conscious state; CRS-R, Coma Recovery Scale-Revised. CRS-R subscales: A, auditory function; Vi, visual function; M, motor function; Ve, verbal; C, communication; Ar, arousal. T0, before the experiment; T1, after a single HD-tDCS session; T2, after treatment for 5 days; #, low EEG signal quality.

## Data Availability

The data that support the findings of this study are available from the corresponding author upon reasonable request.

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
