# Peer review of "Functional Connectivity Increases in Response to High-Definition Transcranial Direct Current Stimulation in Patients with Chronic Disorder of Consciousness"

_brainsci, 2022, doi:10.3390/brainsci12081095_

Round 1

Reviewer 1 Report

Twenty-two patients with disorders of consciousness (DOC) received 10 sessions of high-definition transcranial direct current stimulation (HD-tDCS) with anodal stimulations over left dorsolateral prefrontal cortex (DLPFC). Resting-state EEG data were collected at baseline (T0), after a single session of tDCS (T1), and end-of-treatment (T2). Coma Recovery Scale-Revised (CRS-R) scores were measured at each of those time points. Patients were divided into response group (RE) and non-response group (N-RE) based on whether they have improved CRS-R scores at T2 compared with T0. The authors found increase of CRS-R scores after the HD-tDCS stimulation in the overall sample. They also found the phase-locking value (PLV) at theta band was increased in the left-frontal-parietooccipital region in the RE group while decreased in the N-RE group. The changes of those theta-band PLVs were positively associated with the changes of CRS-R scores. The PLV at alpha-band in the intra-bifrontal region was increased in the RE group and decreased in the N-RE group by T2.

Exploring the effectiveness of using HD-tDCS for treating patients with DOC and understanding its electrophysiological underpinnings are of clinical importance. Although patients were divided in to two groups, all patients received real tDCS and there was no sham control group. This is a major design flaw. There are also other major issues.  My comments are as follows:

Major comments:

1, One major issue of the present study is the lack of sham group. Because of this, the findings reported here can be interpreted as non HD-tDCS effects, such as Time. For example, even though the authors found significant associations between change of theta PLV and changes of CRS-R scores, one can easily argue that neither the change of PLV nor the change of CRS-R scores were caused by HD-tDCS. This does not necessarily mean the current findings are of less importance, but the interpretation of those findings needs to be cautious. It’s recommended discussing the limitation of the study design and its impacts on the explanations of the current findings.

2, The rationale for current data analysis was not clear. For example, why PLV was obtained only at theta and alpha bands? What about the beta and gamma bands? Was PLV calculated based on the electrode-pairs (e.g., F3-P3) or region-to-region (e.g., left frontal – right frontal)?

3, Could the authors provide more details on which electrode-pairs represent left-frontal-parietooccipital region and intra-bifrontal region, respectively? Are those two regions selected based on the significant findings? If so, what are the other regions that were not significant? What’s the method to control for multiple comparison issues? These are critical as the findings here were reported as PLV in those regions rather than electrode-pairs.

Minor comments:

What are the 19 channels?

The notch filter (48-52Hz) seems beyond the band pass frequencies (0.1-40Hz). Why bother use the notch filter?

The number of electrodes (i.e., 19) seems too few for ICA method, and the artifacts rejection threshold (+-150uV) seems to high. Could the authors provide citations for those parameter selections?

Could the authors clarify that the results reported in Figure 3 (i.e., the 2nd paragraph of section 3.1) are from the overall sample (n=22), only the responsive group, or the non-responsive group? It seems from the overall sample, but it was introduced after separating the 2 groups. Maybe consider moving this paragraph before dividing the 2 groups?

The 3-month follow up should be mentioned in the Method section first.

In Figure 6 and 7, it is not clear why reporting a particular patient (i.e., VS patient, NO.8) beside RE and N-RE groups?

Reviewer 2 Report

The stufy by Han et al. demonstrated clinical effects after 10 sessions of HD-tDCS in VS and MCS patients and changes of EEG-derived functional connectivity.  This paper complements previous work of the authors (https://doi.org/10.3389/fnhum.2022.889023) apparently made on the same patient population, in which they demonstrated changes of EEG power spectral density. The study seems robust and general conclusion is sound. Several minor corrections might improve the presentation. Table 1 should be corrected to indicate distribution of patients into RE and N-RE groups. Results of 3 months follow-up might be complemented by CRS-R scores, and more detailed analysis of correlation between EEG data and clinical outcome might therefore be presented. Inability to account for possible spontaneous recovery (especially in early terms) might be considered as a limitation of the study.

Reviewer 3 Report

  1. typo issue, i.e., tenty
  2. Introduction: The use of tDCS devices can enhance patient interaction with the environment, but the effect of tDCS on complex electroencephalographic (EEG) measurements is unclear.   The authors need to explain more about the advantage of their study. A simple search from google scholoar with the key words “tDCS” and “EEG” yield  more than20k results.
  3. DLPFC seems to be the key region, as reported by previous studies. How about other regions? Did all the other regions show null finding when performing tDCS treatment? 
  4. Method: The range of the duration of consciousness impairment seems to be pretty large. How would you control the effect of long-term and short term consciousness impairment (i.e., 302 days, and 29 days).
  5. Etiology of patients is also very different. I would assume patients of differnt etiology group would have different type of brain damage. Why could they be treated with same DLPFC tDCS? 
  6. How many raters participated into the evaluation of CRS-R scores? How did the authors handle the potential rater bias
  7. Trauma seems to be RE-specific, and fluminant encephalomyelitis seems to be N-RE specific. Does this indicate any biological implications?
  8. It is important the highlight the biological meaning of the EEG frequency band in the method section. Besides, why not also analyze delta band?

Round 2

Reviewer 1 Report

The authors have addressed my concerns. I have no further comments and recommend the publication of this manuscript.

Author Response

On behalf of all authors, we would like to express our gratitude to the reviewers for their hard work and for their pertinent suggestions, which have further improved our manuscript.

Reviewer 3 Report

The authors have tried to resolve my comments. This study does have some fundamental limitations, such as no control group, large duration of consciousness, and various etiology types. However, these have been mentioned in the limitations. 

Author Response

First of all, on behalf of all the authors, I would like to thank the reviewers for their hard work. All the valuable comments made by the reviewers have been fully considered. At the same time, we also realize that although some preliminary research results have been obtained, there are still some limitations and shortcomings in this study, which will be actively improved in the future research design to provide more useful evidence for EEG research on consciousness disorders.